# The Reduced Longitudinal Growth Induced by Overexpression of *pPLAIII**γ* Is Regulated by Genes Encoding Microtubule-Associated Proteins

**DOI:** 10.3390/plants10122615

**Published:** 2021-11-28

**Authors:** Jin Hoon Jang, Hae Seong Seo, Ok Ran Lee

**Affiliations:** 1Department of Applied Plant Science, College of Agriculture and Life Science, Chonnam National University, Gwangju 61186, Korea; Jinhun92@naver.com (J.H.J.); ss4540@naver.com (H.S.S.); 2AgriBio Institute of Climate Change Management, Chonnam National University, Gwangju 61186, Korea; 3Interdisciplinary Program in IT-Bio Convergence System, Chonnam National University, Gwangju 61186, Korea

**Keywords:** arabidopsis, phospholipase, anisotropic, longitudinal, microtubule-associated protein

## Abstract

There are three subfamilies of patatin-related phospholipase A (pPLA) group of genes: *pPLAI*, *pPLAII*, and *pPLAIII*. Among the four members of *pPLAIIIs* (*α*, *β*, *γ*, *δ*), the overexpression of three isoforms (*α*, *β*, and *δ*) displayed distinct morphological growth patterns, in which the anisotropic cell expansion was disrupted. Here, the least studied *p**PLAIII**γ* was characterized, and it was found that the overexpression of *p**PLAIII**γ* in Arabidopsis resulted in longitudinally reduced cell expansion patterns, which are consistent with the general phenotype induced by *pPLAIII*s overexpression. The microtubule-associated protein MAP18 was found to be enriched in a *pPLAIIIδ* overexpressing line in a previous study. This indicates that factors, such as microtubules and ethylene biosynthesis, are involved in determining the radial cell expansion patterns. Microtubules have long been recognized to possess functional key roles in the processes of plant cells, including cell division, growth, and development, whereas ethylene treatment was reported to induce the reorientation of microtubules. Thus, the possible links between the altered anisotropic cell expansion and microtubules were studied. Our analysis revealed changes in the transcriptional levels of microtubule-associated genes, as well as phospholipase D (*PLD*) genes, upon the overexpression of *p**PLAIII**γ.* Overall, our results suggest that the longitudinally reduced cell expansion observed in *pPLAIII**γ* overexpression is driven by microtubules via transcriptional modulation of the *PLD* and *MAP* genes. The altered transcripts of the genes involved in ethylene-biosynthesis in *p**PLAIII**γOE* further support the conclusion that the typical phenotype is derived from the link with microtubules.

## 1. Introduction

The whole architecture of plant growth and development depends on anisotropic cell expansion [1]. The driving force of this cell expansion is provided by turgor pressure, while cell wall components direct cell expansion. Cellulose microfibrils in the cell wall and cortical microtubules have long been considered as the important factors for the anisotropic growth of plant cells [1]. However, it is also noticeable that, although the disruption of transverse microtubule arrays patterns in elongating cells leads to isotropic plant cell expansion, this is not always necessarily accompanied by the disruption of transverse microfibrils [2]. These reports suggest that other cell wall components could also be involved in directing cell expansion. Phospholipase acts on the plasma membrane, where links between the microtubules and microfibrils have been suggested to be one of the possible key players for the communication between the two structural components [2].

Phospholipases A (PLAs) are a multigene family of enzymes in plants. PLAs are subdivided into PLA_1_s, secretory PLA_2_s (sPLA_2_s), and patatin-related PLAs (pPLAs) [3,4,5]. pPLAs have been shown to induce altered anisotropic cell expansion when overexpressed in Arabidopsis [5,6,7,8,9,10], rice [11,12], camelina [13], and poplar [14,15]. Although isotropic cell expansion patterns and occasional reduced longitudinal growth have been observed in previous studies, the underlying mechanism was first reported in the study of *pPLAIIIβ* overexpression (*pPLAIIIβOE*) in Arabidopsis [7]; *pPLAIIIβOE* resulted in decreased cellulose deposition and lipid remodeling, which led to a reduced longitudinal cell elongation [7]. As cellulose deposition is tightly linked with microtubule arrays, it was proposed that *pPLAIIIβOE-*mediated cell expansion patterns might be linked to microtubule dynamics. This proposition was substantiated by another *pPLAIIIδ* overexpression study that also displayed inhibited longitudinal growth but promoted transverse growth in Arabidopsis and *Brassica napus* [9]. Proteomic analysis of *pPLAIIIδOE* plants revealed enriched microtubule-associated protein MAP18 that modulates directional cell expansion [16]. Overexpression of other closely related isoforms, *pPLAIII**α* with 72% identity to *pPLAIIIβ,* and *PgpPLAIIIβ* (*pPLAIIIβ* homolog isolated from ginseng) with an average 61% identity to *pPLAIII**α* and *pPLAIIIβ*, isolated from Arabidopsis [10,14], also resulted in defective anisotropic cell expansion. However, only the content and deposition of lignin, not cellulose, were reduced [10,14,15]. We hypothesized that the changes in the composition and content of lignin caused by phospholipase enzymes could also alter the arrays of microtubule structure by directly or indirectly altering cellulose structure.

Microtubules play a crucial role in directional cell expansion [17,18] via interaction with microfibrils and other unidentified cell wall components. Thus, identifying the regulatory factors of the cortical microtubule cytoskeleton is crucial in understanding the anisotropic cell growth patterns. Overall, in the floral organs and leaves, anisotropic cell expansion is accompanied by overexpression of *p**PLAIII**γ*, which is the least characterized among all *pPLAIIIs*. Quantification analysis were carried out for the transcripts of genes encoding phospholipase Ds (*PLDs*) [19], microtubule-associated proteins [20], and those involved in ethylene biosynthesis [21,22], which are known to regulate microtubules. Altogether, results suggest that the transversely expanded cell elongation is caused by changes in the *PLD*-mediated expression of microtubule-associated genes.

## 2. Results

### 2.1. Overexpression of pPLAIIIγ Affects Anisotropic Cell Elongation

Overall, stunted growth patterns with reduced plant height and altered anisotropic cell expansion observed in stem, root, leaf, and floral organs are general phenotypes in three independent overexpression lines of *pPLAIIIα* [5], *pPLAIIIβ* [7], and *pPLAIIIδ* [9], among four isoforms of *pPLAIII* genes *(pPLAIIIα*, *-β*, *-γ*, *-δ)* identified in Arabidopsis [23,24]. Only the phenotypic characteristics of the *pPLAIIIγ* (*pPLAIIIγOE*) in each part of organs have not yet been characterized in Arabidopsis. A T-DNA insertional mutant (SALK_088404) has been reported by the same group to have a negligible level of transcripts as determined via quantitative real-time polymerase chain reaction (qRT-PCR) [25,26]. However, in our study, the transcripts were not significantly reduced in the SALK line (Appendix A), and an additionally analyzed line (SAIL_832_E01) showed only 35% reduced transcripts compared to the wild type (Appendix A). Due to the lack of knock-out mutant, only the overexpression line was characterized in this study.

To evaluate whether the *pPLAIIIγOE* also displays similar dwarf plant growth patterns, the full-length (1690 bp; upstream 6 bp was added from the start) genomic DNA sequence of Arabidopsis *p**PLAIII**γ* was overexpressed under 35S promoter with yellow fluorescence protein (YFP) tagging at the C-terminal end. Three independent transgenic lines that showed ectopic overexpression of *p**PLAIII**γ* (Figure 1A) were chosen for further characterization, following Mendelian genetic segregation. Stunted dwarf plant height of *pPLAIIIγOE* focused on stem elongation and xylem lignification is reported separately [27]. This study focuses on cell elongation phenotypes in floral organs and leaves. Floral organs were shorter and more rounded in the OE lines than in the Col-0 control (Figure 1B). Scanning electron microscopy (SEM) image further confirmed the radially expanded and longitudinal reduction of cell elongation patterns in the whole flower, including ovary, filament, and petal (Figure 1C). A longitudinally inhibited growth pattern was also observed in *p**PLAIII**γOE* pollen (Figure 1C). Overall, the observed phenotypes are very similar to those of *pPLAIIIαOE* [5], *pPLAIIIβOE* [7], and *pPLAIIIδOE* [9]. However, it is noticeable that the pollen displayed a more severely altered morphological structure compared with that of *pPLAIIIαOE* [5]. Thus, pollen tube growth was further measured and showed that the pollen tube length was much reduced (Figure 1D,E).

### 2.2. YFP-Tagged pPLAIIIγ Is Localized to the Plasma Membrane

To determine the subcellular localization of pPLAIIIγ, stable transformants of C-terminal yellow fluorescence protein (YFP)-tagged pPLAIIIγ were imaged using root cells of 4-days grown seedlings. All transgenic plants showed plasma membrane (PM) localization perfectly merged with FM4-64 (Figure 2A). To verify that the PM localization is not associated with the cell walls, plasmolysis by treating seedlings with 0.2 M NaCl was induced. After plasmolysis, the YFP signal was separated from the cell wall with the protoplast, indicating that pPLAIIIγ is not cell wall-associated (Figure 2B).

### 2.3. pPLAIIIγOE Displayed Altered Anisotropic Cell Expansion in the Leaf and Trichome

Leaf surface areas were reduced by 46% and 95% in the intermediately overexpressing line #5 and strongly expressing line #14 (Figure 3A,B), respectively. Corresponding to the surface area, leaf fresh weight was also reduced. However, the water content and thickness of leaves were increased in the OE lines (Figure 3B). The reduced leaf surface area and fresh weight were derived from radially expanding cell growth patterns rather than longitudinal growth in a rosette and cauline leaf. SEM images further showed round-shaped cell growth patterns in the adaxial and abaxial sides of leaves (Figure 3C). To clearly illustrate the radial rather than longitudinal expansion pattern of epidermis cells, exact cell boundaries were highlighted (Figure 3D), and the ratio of longitudinal to transverse axes on adaxial epidermis of rosette leaves (Figure 3E) was estimated. The ratio was 2.9 in Col-0, whereas it was 1.4 in the OE line, which indicated a 52% reduction in the longitudinal/transverse axis ratio (Figure 3E). Two or four-branched trichomes were also observed in the OE line compared to those in the three-branched wild-type.

### 2.4. pPLAIIIγOE Increased Seed Size

As expected from the flower organ size (Figure 1B,C), silique length was decreased in *p**PLAIII**γOE* (Figure 4A). The overall seed size was enlarged in two strong OE lines, #1 and #14 (Figure 4B). The exact measurement of the length and width of OE seeds displayed that their cross-sectional elongation was increased, with no effect on their longitudinal elongation pattern (Figure 4C). Accordingly, the seeds were more rounded and enlarged. Regardless of the seed size change, the germination rate was not altered (Figure 4D) compared to that with *pPLAIIIαOE* [5].

### 2.5. PLD Genes Are Downregulated by pPLAIIIγOE

Transcriptional modulation of *PLD* genes were observed in previous study of *pPLAIIIαOE* [5]. To verify the links between a plasma membrane-localized pPLAIIIγ with *PLD* genes, a quantitative gene expression study was performed. Four major isoforms of *PLD* genes, including *PLDα1*, *PLDδ*, *PLDζ1*, and *PLDζ2*, were significantly downregulated in 3-week-grown rosette leaves of *p**PLAIII**γOE* lines (Figure 5A). The modulation of *PLD**α1* and *PLDζ1* transcripts was less significant in the stem of *p**PLAIII**γOE* lines, but the mRNA levels of *PLD**δ* and *PLDζ2* were still downregulated (Figure 5B). *PLDζ2* was shown to be implicated in the trafficking of the PIN-FORMED2 (PIN2) auxin-transporting plasma membrane (PM) protein [28]. Thus, it is also an intriguing part to explore whether the overexpression of *p**PLAIII**γ* has a function in the regulation of the auxin transporters as reported previously in related work [3]. For anisotropic cell expansion, downregulated *PLD**δ* gene expression in both leaf and stem is more intriguing since PLDδ is reported to bind to microtubules [19]. Thus, *pPLAIII*-mediated anisotropic cell expansion seems to result from the modulation of microtubule-binding PLDδ.

### 2.6. Genes Encoding Microtubule-Associated Proteins Are Modulated by pPLAIIIγOE

Microtubules influence plant organ morphology. The proteins that interact with microtubules mediate their functional and structural interaction with other cell structures that ultimately modulate microtubule dynamics and organization [29]. The *p**PLAIII**γ*-mediated reduced expression of *PLD* genes led us to evaluate the possible modulation of several reported microtubule-associated genes encoding MAP proteins, including MAP18, identified by proteomic analysis as being enriched in *pPLAIIIδOE* transgenic plants [16]. In this study, we observed that *MAP18* and *MAP20* were significantly downregulated, while *MAP65-1* and *MAP70-1* were significantly upregulated in 3-weeks grown leaves (Figure 6A). The mRNA level changes were less significant in the stem (Figure 6B), where only *MAP20* and *MAP70-5* were up- and downregulated, respectively. The rest of the *MAP* genes were not altered (Figure 6B).

### 2.7. Ethylene Biosynthesis Genes Are Modulated by pPLAIIIγOE

Our previous study showed that the overexpression of *pPLAIIIα* altered gene expression involved in ethylene biosynthesis during seed germination [5] and the seedling stage [15]. To evaluate whether the altered gene expression involved in ethylene-biosynthesis is a general feature of *pPLAIIIs* overexpression, a quantitative gene expression study was performed with three ACC (1-aminocyclopropane-1-carboxylic acid) synthase (*ACS*) and ACC oxidase (*ACO*) genes (Figure 7). Except for the mRNA level of *ACS11*, which was downregulated, all of the tested genes were upregulated upon *p**PLAIII**γOE* (Figure 7). In the ethylene biosynthesis pathway, the ethylene precursor ACC is synthesized by the *ACS* gene and, finally, oxidized by *ACO* genes to produce ethylene. Thus, the increased transcripts of all the three major *ACO* genes, *ACO1*, *ACO2*, and *ACO4*, indicate that *p**PLAIII**γOE* also affects ethylene biosynthesis in leaves.

## 3. Discussion

The structural changes in the cell wall and their correlation with anisotropic cell expansion patterns were monitored by overexpression of *pPLAIIIβ* [7,10,14] and *pPLAIIIα* [5,15]. Subcellular links of microtubule-associated proteins with *pPLAIIIs*-associated cell wall components were examined for further clarification. To reveal the molecular link between microtubule-associated proteins and patatin-related phospholipase A genes, overexpression lines of the *p**PLAIII**γ* gene in Arabidopsis were generated, and relevant changes in transcript levels were quantified. Due to the lack of available knock-out mutant lines (Appendix A) in the stock center, the mutant was not analyzed. As observed in the overexpression lines of other close homologs (*pPLAIIIα-*, *pPLAIIIβ-*, and *pPLAIIIδOE*) [5,6,7,8,9,10,12,13,14,15], *p**PLAIII**γOE* also showed a longitudinally reduced growth pattern in flower organs and leaves (Figure 1 and Figure 3). *p**PLAIII**γOE* displayed an overall radially expanded cell growth pattern that was defective in anisotropic cell expansion. Isotropic cell expansion was more apparent in the seeds of *p**PLAIII**γOE* lines (Figure 4), with no changes in the seed germination rate. The consistent germination rate observed in *p**PLAIII**γOE* lines was distinct compared to that in *pPLAIIIαOE,* which showed an initially increased germination rate [5]. Subcellular plasma membrane localization using tagged fluorescent protein was reported for pPLAIIIα, pPLAIIIβ, and pPLAIIIδ [5,7,25]. C-terminal YFP-tagging with pPLAIIIγ also showed apparent localization in the plasma membrane, indicating its function in the subcellular domain via interaction with other cellular components for altered anisotropic growth.

Anisotropic cell expansion plays a pivotal role in plant growth and development, and it can be regulated by cell wall components that interact with the oriented deposition of cellulose microfibrils [1]. Among several major structural components of cell walls, only cellulose has been well characterized relative to the cell growth direction within native tissue. However, lignin molecules have also been reported to be involved in determining the geometry of the cell walls by showing that the aromatic rings of the phenyl propane structural units are parallel to the plane of the surface of the cell wall [30]. In *pPLAIIIβOE* transgenic lines, cellulose content was decreased [7]. However, only lignin content was decreased in *pPLAIIIαOE* lines, and cellulose content remained unaltered [10,14,15,27]. This suggests that lignin molecules are also involved in anisotropic cell expansions.

Phospholipids are the key structural components of plasma membranes and signaling cascades. They are conserved across in a wide range of species in different proportions, with conversion processes that involve hydrophilic enzymes, such as phospholipase-C (PLC), phospholipase-D (PLD), and phospholipase-A (PLA). Phospholipase Dδ (PLDδ) was identified by screening an Arabidopsis cDNA expression library with monoclonal antibody 6G5 against the tobacco 90-kD polypeptide (p90) that was found in microsomal factions and colocalized with cortical microtubules [19,31]. Cell growth direction usually depends on the organization of cortical microtubule arrays [32]. This suggests that PLD is involved in relaying signals to the microtubule cytoskeleton. Considering the potential crosstalk between the phosphatidic acid-producing PLD and lysophospholipid-producing PLA pathways in the plant-bacterial pathogen interaction [33], the *PLAIII*-mediated regulatory roles of *PLDs* are supported strongly. Transcriptional modulation of *PLD* genes in *pPLAIIIαOE* [5] further substantiates this notion. Through the interaction with microtubules, the PM-localized PLDδ is activated, which leads to the release and reorientation of microtubules [34]. PLDδ is, thus, considered a target protein potentially integrating multiple structures into a functional complex in plants [29]. Overall, significantly downregulated transcripts of *PLD**δ* due to overexpression of *p**PLAIII**γ* in the leaf and stem (Figure 5) strongly suggest that there are *PLD*-mediated modulations, wherein genes encoding microtubule-associated protein might be regulated. Complementary to this theory, the expression of three major isoforms of *PLD* genes, namely *PLDα1*, *PLDζ1*, and *PLDζ2*, was also significantly decreased upon the overexpression of *pPLAIIIα* [5]. Therefore, transcriptional changes in several genes encoding microtubule-associated proteins [20] could offer positive clues. MAP18 and MAP70-5 destabilize and stabilize microtubules, respectively [35,36]. MAP65-1 is a microtubule crosslinker, and MAP70-1 functions in microtubule assembly [36,37]. Overexpression of *MAP70-5* induces right-handed helical growth [36] and that of *MAP20* results in helical cell twisting [38]. Upon constitutive overexpression of *p**PLAIII**γ*, the mRNA levels of all the tested *MAP* genes, except those of *MAP70-5*, were altered in the leaf; *MAP70-5* was found to be slightly downregulated in the stem (Figure 6). Interestingly, the expression of gene encoding *MAP18* that was enriched by the overexpression of *pPLAIIIδ* [16] was reduced by 41% (Figure 5). Considering the synergistic or antagonistic modulations among *pPLAIII* genes [5], further studies focusing on the crosstalk of *p**PLAIII**γ* and *pPLAIIIδ* would seem worth undertaking.

The altered transcripts involved in ethylene biosynthesis are also important factors regulating the anisotropic growth pattern. Not only *ACO2*, but also *ACO1* and *ACO4* that are known to exhibit ACC oxidase activity in vitro [39], were all upregulated (Figure 7). Although these results are similar to a those of a previous study [5], the underlying mechanism focused on the longitudinally reduced and transversely expanded cell elongation patterns still remains elusive. Nevertheless, ethylene treatment induced the reorientation of both microtubules and newly deposited microfibrils from transverse to the longitudinal direction [40]. Thus, microtubules regulated via the modulation of *PLD* genes are possibly crucial for the anisotropic expansion in *p**PLAIII**γOE* plants (Figure 8). The microtubule-nucleating protein tubulin could be a link in this interaction via its hydrophobic domain or an indirect interaction with an integral membrane protein [41]. Plasma-membrane-located phospholipase could be the link for the interaction with microtubules. Altogether, these results suggest that isotropic cell expansion is associated with the *PLD*-mediated alteration of *MAP* genes (Figure 8).

## 4. Materials and Methods

### 4.1. Plant Materials and Growth Conditions

Columbia ecotype (Col-0) of *Arabidopsis thaliana* was used as the background for all wild-type, knockout mutants, and overexpression lines in this study. The *pplaIIIγ* knockout mutants (*pplaIIIγ#1*: SALK_088404 and *pplaIIIγ#2*: SAIL_832_E01) were purchased from the Arabidopsis stock center (http://www.arabidopsis.org/, (accessed on 25 June 2021)). Arabidopsis seeds were sown on half-strength MS medium (Duchefa Biochemie, Haarlem, The Netherlands) containing 1% sucrose, 0.5 g/L of 2-[N-morpholino] ethanesulfonic acid (MES), and 0.8% phytoagar; the pH was adjusted to 5.7. After 2 days of vernalization in dark at 4 °C, imbibed seeds were grown under long-day light conditions (16 h light/8 h dark) at 23 °C. Germinated seedlings were transferred to an autoclaved soil mixture containing soil, vermiculite, and pearlite (3:2:1 *v*/*v*/*v*).

### 4.2. Transgenic Construct and Arabidopsis Transformation

To overexpress *pPLAIIIγ*, the full-length genomic sequence of *pPLAIIIγ* was obtained by PCR using Arabidopsis wild-type (Col-0) genomic DNA as a template and primers listed in Appendix A. The *pPLAIIIγ* was cloned into the modified pCAMBIA1390 vector under the control of the 35S cauliflower mosaic virus promoter with yellow fluorescence protein (YFP) C-terminal tagging. The recombinant construct was confirmed by sequencing and introduced into *Agrobacterium tumefaciens* C58C1 (pMP90). Arabidopsis plants were transformed using the floral dipping method. Transformants were selected on a half-strength MS medium with hygromycin (50 μg/mL) and confirmed by PCR. Finally, three independent homozygous lines were used for further experiments.

### 4.3. qRT-PCR

Total RNA was extracted from different organs of Arabidopsis using Pure link™ RNA Mini Kit (Invitrogen, Carlsbad, CA, USA) with on-column DNase I treatment to remove residual genomic DNA. Extracted total RNA was verified for quality and quantity using a Nano-MD UV-Vis spectrophotometer (Scinco, Seoul, Korea). The complementary DNA (cDNA) was synthesized in a total 20 μL reaction volume using RevertAid Reverse transcriptase (Thermo, Waltham, MA, USA). qRT-PCR was performed using TB Green™ Premix Ex Taq™ (Takara, Shiga, Japan) and Thermal Cycle Dice real-time PCR system (Takara, Shiga, Japan), as previously reported [10]. Target gene-specific primers for the qRT-PCR analysis are listed in Appendix A.

### 4.4. Pollen Germination and Pollen Tube Length Measurement

Pollen grains were incubated in the pollen germination medium (PGM). A modified PGM based on that of Reference [42] was used. In brief, it contained 0.01% H_3_BO_3_, 1 mM CaNO_3_, 1 mM MgSO_4_, 1 mM CaCl_2_, 500 mg/L MES (Duchefa Biochemie, The Netherlands), 10 mg/L Myo-inositol (DAEJUNG, Korea), 1.8% sucrose (Duchefa Biochemie, The Netherlands), and 0.75% agarose. The PGM was adjusted to pH 7.3 with crystal Tris (Affymetrix, Santa Clara, CA, USA). Following incubation at 23 °C for 6 h in the dark, germinating pollen grains were observed and photographed using a microscope (DM3000 LED, Leica, Wetzlar, Germany). The pollen tube length was measured using ImageJ software (imagej.nih.gov/ij/index.html, Wisconsin, USA) at 1st of September 2021.

### 4.5. Observation of Reporter Gene Expression

The fluorescence signal from the reporter protein and the FM4-64 dye (70021, Biotium, Fremont, CA, USA) was observed using confocal laser scanning microscopy (TCS SP5 AOBS Tandem, Leica, Germany). Four-day-old seedlings were stained with 2 μM FM4-64 for 5 min, washed with distilled water for 5 min, and plasmolyzed by 0.2 M NaCl treatment for 1 min. YFP and RFP signals were detected using 514/>530 and 543/560–615 nm excitation/emission filter sets, respectively. Fluorescence images were digitized using the Leica LAS X program. The images were acquired at the Korea Basic Science Institute, Gwangju, Korea.

### 4.6. SEM Analysis

Rosette leaves, cauline leaves, and flowers were used for SEM analysis to compare the surface tissue phenotypes of Col-0 and *pPLAIIIγ*OE#14. All surface images were obtained using a low-vacuum scanning electron microscope (model no. JSM-IT300; JEOL, Korea) at a 10.8-mm working distance and 20.0 kV.

## Figures and Tables

**Figure 1 plants-10-02615-f001:**
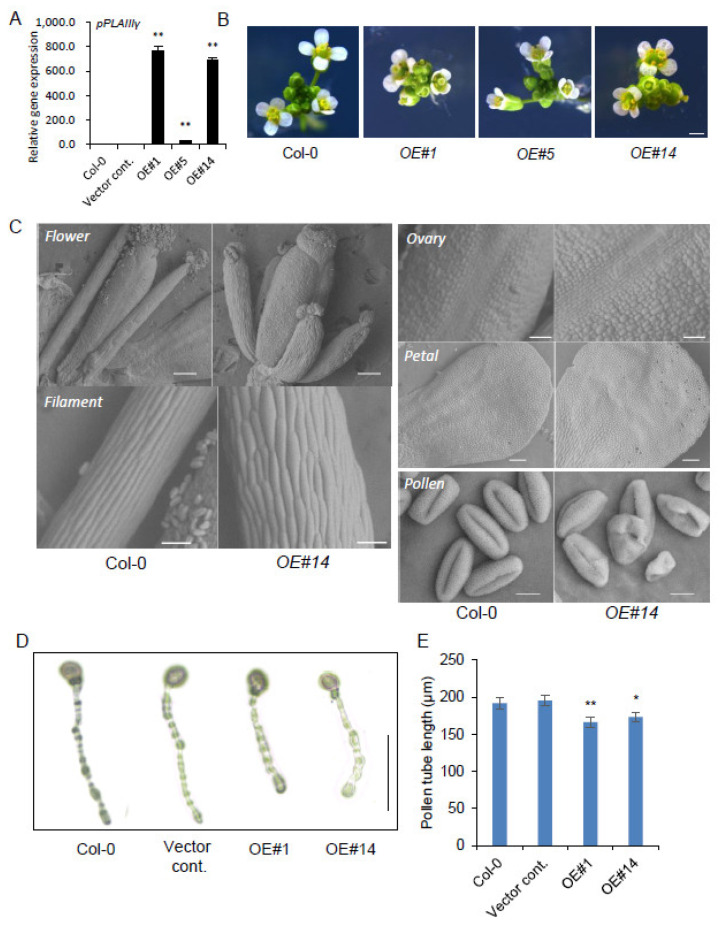
Overexpression of *pPLAIIIγ* altered cell elongation patterns in floral organs. (**A**) The expression level of *pPLAIIIγ* in the 2-week-old seedling. Data represent the average ± standard error (SE) from three independent replicates at *p <* 0.01 (**). (**B**) Flower phenotypes of Col-0 and *pPLAIIIγOEs.* (**C**) Scanning electron microscopy (SEM) image of floral organs from Col-0 and *pPLAIIIγ**OEs*. Scale bars = 1 mm (B), 200 μm (C—flower), 100 μm (C—petal), 50 μm (C—style and filament), and 10 μm (C—pollen). (**D**) Pollen tube morphology of controls and *OEs* after in vitro pollen incubation for 6 h. Scale bar = 100 μm. (**E**) Pollen tube length of controls and *OEs*. Data represent the average ± SE from independent replicates at *p <* 0.05 (*) and *p <* 0.01 (**). *n* = 232 (Col-0), 236 (vector cont.), 195 (OE#1), and 222 (OE#14).

**Figure 2 plants-10-02615-f002:**
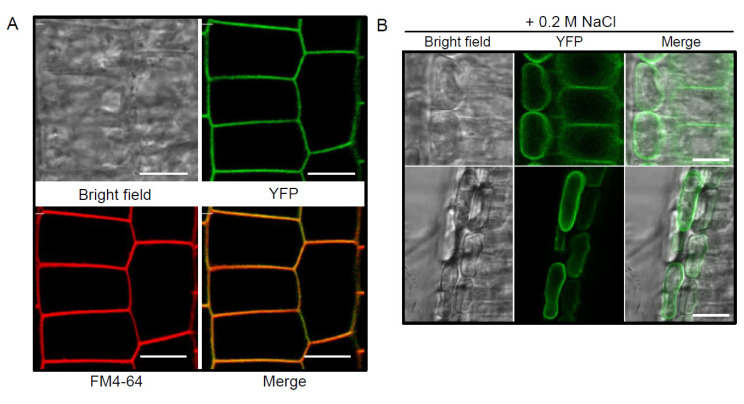
Subcellular localization of pPLAIIIγ-YFP in the plasma membrane. (**A**) Fluorescence image from the primary root of 4-day-old pPLAIIIγ-YFP seedling merged with FM4-64. (**B**) Plasmolysis of root epidermal cells of the pPLAIIIγ-YFP treated with 0.2 M of NaCl for 1 min. Scale bars = 100 μm.

**Figure 3 plants-10-02615-f003:**
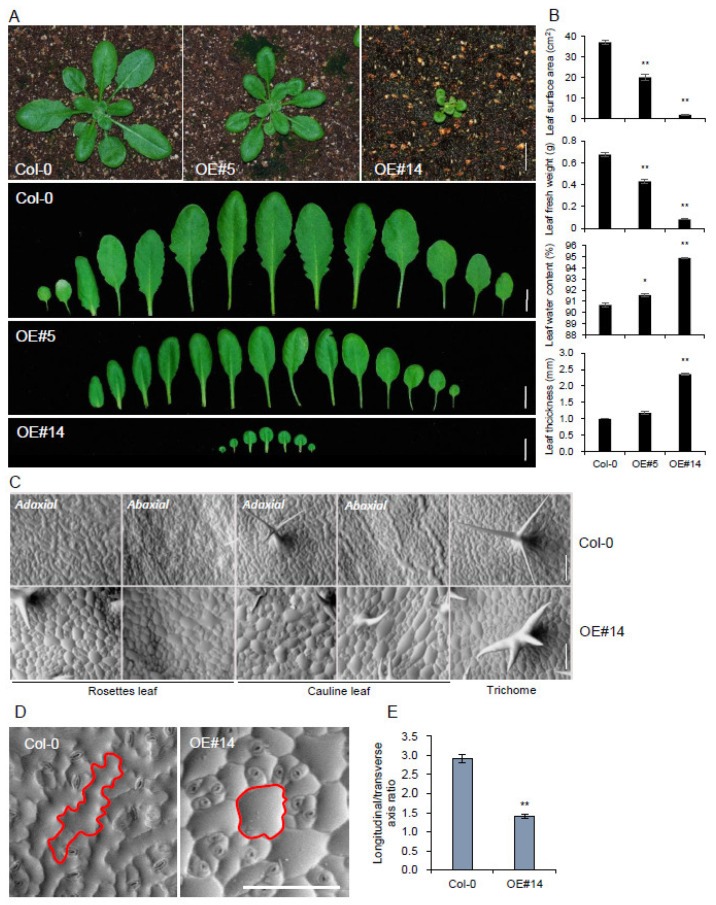
Overexpression of *pPLAIIIγ* alters the size and shape of leaves, including the epidermal cells. (**A**) The aerial part of each 4-week-old plant and individual leaves. The leaves are arranged from cotyledons (**left**) to the youngest leaves (**right**). (**B**) Statistical analysis of leaf surface area, leaf fresh weight, leaf water content, and leaf thickness in 4-week-old plants. (**C**) SEM images of adaxial and abaxial sides of epidermal cells of the Col-0 (**upper lane**) and *pPLAIIIγ**OE*#14 (**lower lane**), respectively. (**D**) Magnified image of epidermal cells of the Col-0 and *pPLAIIIγOE*#14. Red lines highlighted a single epidermal cell. (**E**) The ration of longitudinal to transverse axes of epidermal cells on adaxial side of rosette leaf. Scale bars = 1 cm (**A**) and 100 μm (**C**,**D**). Data represent the means ± SE of multiple replicates. *n* = 3 (**B**), *n* = 40 (**E**). Asterisks indicate significant difference using Student’s *t*-test (* *p* < 0.05 and ** *p* < 0.01) compared with the Col-0.

**Figure 4 plants-10-02615-f004:**
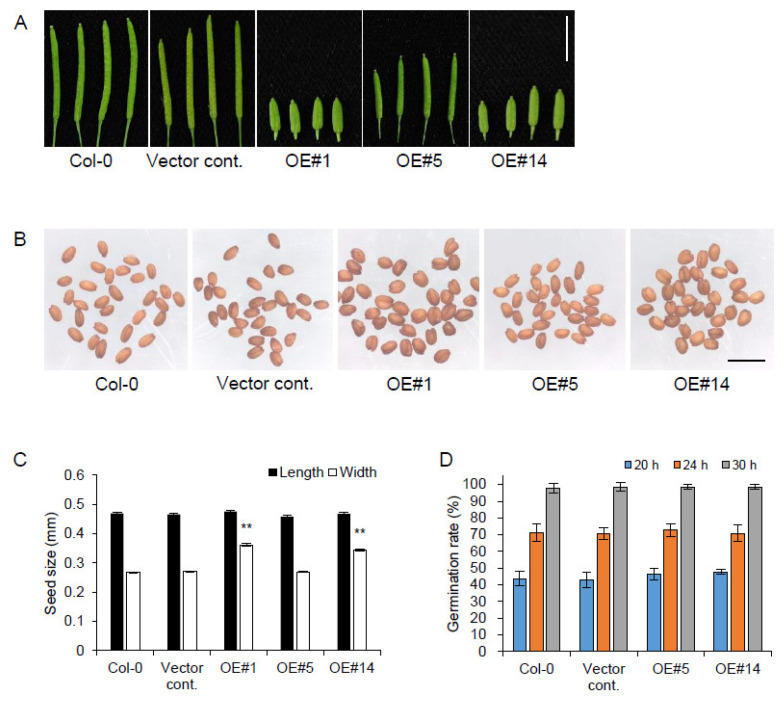
Seed width was increased in *pPLAIIIγ**OE*. (**A**–**C**) Siliques (**A**), mature seeds (**B**), and seed size (**C**) of controls, and *pPLAIIIγ**OE* lines. (**D**) Germination rates of control and *OE* lines after 20-h germination under continuous light conditions. Scale bars = 5 mm (**A**) and 1 mm (**B**). Data represent the average ± SE from independent replicates at *p <* 0.01 (**). *n* = 50 (**C**) and 140 (**D**).

**Figure 5 plants-10-02615-f005:**
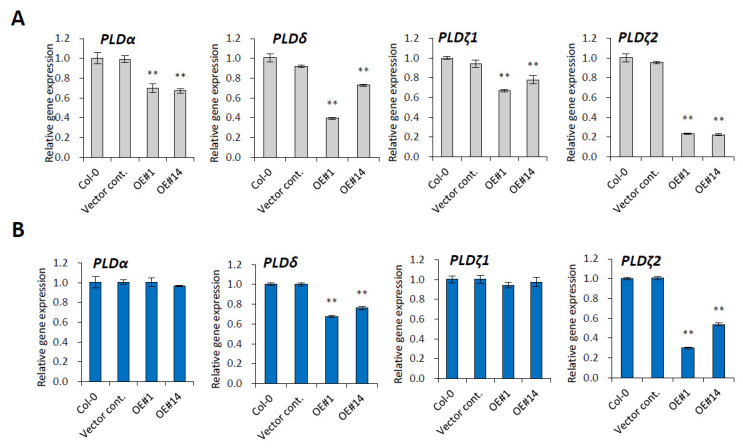
Phospholipase D (*PLD*) was downregulated in *pPLAIIIγ**OE*. Expression levels of *PLD* genes (*PLDα*, *PLDδ*, *PLDζ1*, *PLDζ2*) in (**A**) 3-week-old rosette leaf and (**B**) 7-week-old stem. Data represent the average ± SE from three independent replicates at *p <* 0.01 (**).

**Figure 6 plants-10-02615-f006:**
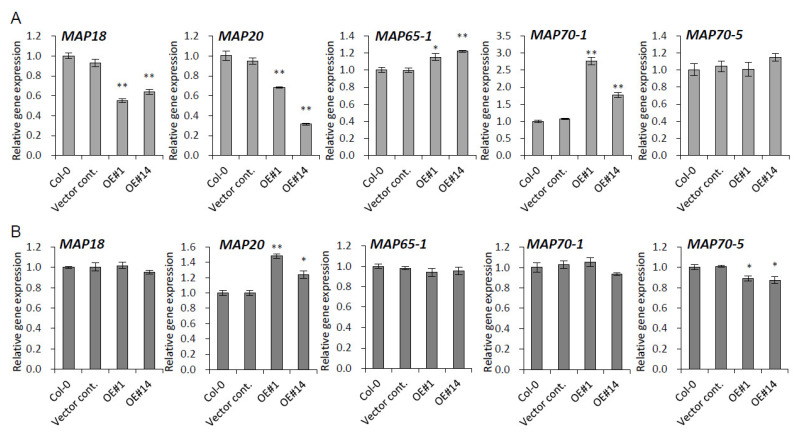
Expression levels of microtubule-associated protein (*MAP*) genes in *pPLAIIIγ**OE*. Expression levels of *MAP* genes in (**A**) 3-week-old rosette leaf, and (**B**) 7-week-old stem. Data represent the average ± SE from three independent replicates at *p <* 0.05 (*) and *p <* 0.01 (**).

**Figure 7 plants-10-02615-f007:**
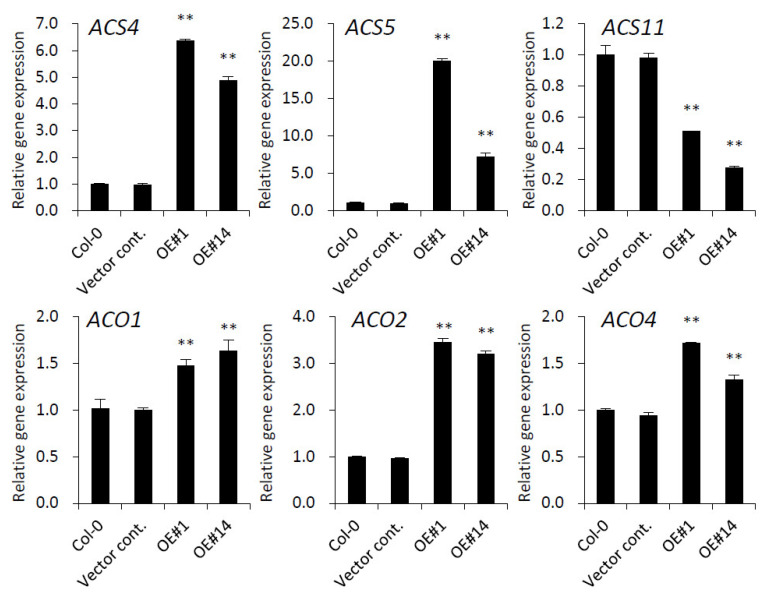
Ethylene biosynthesis-related genes were upregulated in *pPLAIIIγ**OE*. Transcript levels of *ACS* and *ACO* genes in 3-week-old rosette leaves of *pPLAIIIγ**OE*. Data represent the average ± SE from three independent replicates at *p <* 0.01 (**).

**Figure 8 plants-10-02615-f008:**
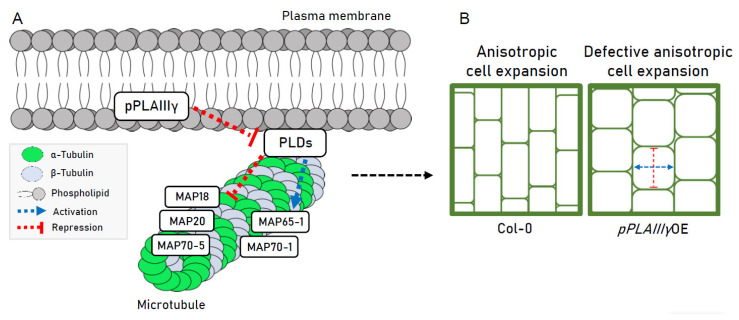
A hypothetical diagram of pPLAIIIγ-PLD-MAP-mediated regulation in an anisotropic cell elongation pattern, based on transcripts modulation by overexpression of *pPLAIIIγ*. (**A**) Overexpression of *pPLAIIIγ* reduces the mRNA levels of *PLD* genes. Reduced expression of *PLD* genes caused transcriptional regulation of *MAPs.* Plasma membrane- associated PLDδ connects to the microtubules and is involved in the release and organization of microtubules [19,34]. MAPs play important roles in microtubule polymerization, stabilization, and bundling. The reported functions of *MAP* genes modulated by pPLAIIIγ-PLD regulation are listed: MAP18 and MAP70-5 are involved in the destabilization and stabilization of microtubules, respectively [35,36]. MAP65-1 is a microtubule crosslinking protein, and MAP70-1 plays a role in microtubule assembly [36,37]. MAP20 is involved in the regulation of plant helical growth [38]. (**B**) Normal anisotropic cell expansion patterns are observed in Col-0 (left panel). This study suggests that the defective anisotropic cell expansion (longitudinally reduced and radially expanded pattern, right panel) by *pPLAIIIγ*OE seems to be affected by pPLAIIIγ-PLD-MAP co-regulation. PLD: phospholipase D, MAP: microtubule-associated protein.

## Data Availability

The data presented in this study are available in Figure 1, Figure 2, Figure 3, Figure 4, Figure 5, Figure 6, Figure 7 and Figure 8 and Appendix A.

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
