# Peer review of "The Reduced Longitudinal Growth Induced by Overexpression of *pPLAIII**γ* Is Regulated by Genes Encoding Microtubule-Associated Proteins"

_plants, 2021, doi:10.3390/plants10122615_

Round 1
Reviewer 1 Report
Please see the review attached.

Author Response
Reviewer 1
I’ve read this paper with great pleasure. It includes many experimental data, which confirm that overexpression of Arabidopsis pPLAIIIγ showed longitudinally reduced patterns of cell expansion, which has clear influence on the general pPLAIIIs plant phenotype in the case of overexpression. The authors unambiguously demonstrated that there were transcriptional changes not only of the genes connected with microtubules but also phospholipase D (PLD) genes due to pPLAIIIγ overexpression. The paper also shows that longitudinally reduced cell expansion observed in pPLAIIIγ overexpression is influenced by microtubules through transcriptional modulation of PLD and MAP genes. Furthermore, the authors are right in stating that the changed transcript taking part in ethylene biosynthesis in pPLAIIIγOE additionally makes the hypothesis formulated in this paper more credible – the hypothesis that the typical phenotype most probably originates from the interaction with microtubules.
The authors have described their research process in detail in section “Materials and Methods” (it could be easily replicated). The results were presented with the use of 10 well-compiled diagrams and four complex figures with micrographs. Also, the supplemental materials are useful for readers.
The reviewed work includes an accurately selected literature. I also haven’t noticed any mistakes in citation.
Overall, this is a valuable manuscript, worth publishing.
A list of minor corrections below:
Line 2 – In the paper’s title: “Growth by” - replace with “growth due to”
[Response]: Thanks for your comment. However, ‘by’ seems more right rather than ‘due to’ since we artificially (ectopically) expressed the gene rather than it naturally happened. Because of this reason, the following same requests are not changed except your requested Line 68 and Line 78 issues.
Abstract:
Line 19 – “characterized pPLAIII. characterized and observed”- this fragment is not clear, repetitive phrases convolute the meaning - please rephrase
[Response]: As your comment, it is revised.
‘Here, the least characterized pPLAIIIγ was characterized and observed that~’
Line 30 – it should be: “further support the conclusion that the typical phenotype”
[Response]: As your comment, it is revised.
Introduction:
Line 41 – “not all the time” - wasn't this meant as "not always"? If so, then this is more grammatically correct.
[Response]: As your comment, it is revised.
Line 51 – “which means altered isotropic and/or reduced longitudinal growth” - this fragment should be between commas; also instead of "which means" there should be "i.e."
[Response]: As your comment, it is revised.
Line 62 – “that is” - replace with "which is"
[Response]: As your comment, it is revised.
Line 67 – “by the function” - replace with "due to"/"thanks to" instead of "by"
[Response]: As your comment, it is revised. (by the function à due to the function)
Line 78 – “due” - replace with “is occurring due to”
[Response]: As your comment, it is revised.
Results
Line 96 – “additionally added” - pleonasm, redundant - remove "additionally"
[Response]: As your comment, it is revised.
Lines 104-106 – “confirmed that the radially expanded and longitudinally reduced cell elongation patterns in the whole flower including ovary, filament, and petal” - grammmar (subject-verb) mistake: there is no verb complementary to the last subject of the sentence; UNLESS you meant: "Scanning electron microscopy (SEM) image further confirmed that the radially expanded and longitudinal reduction of cell elongation patterns...."- if this is what you meant, remove "that" and change "reduced" to "reduction"
[Response]: Yes, it means that exactly as you comment. As your comment, it is revised.
Figure 1.
Line115 – “Flower phenotype” replace with “Flower phenotypes”
[Response]: As your comment, it is revised.
Line 118 – instead of "multiple" indicate how many replicates exactly, for precision
[Response]: As your comment, added number of detailed replicates.
Figure 3.
Line 143 – “alters the size and shape of epidermal cells” - replace with ”alters the size and shape of leaves, including i.a. epidermal cells”
[Response]: As your comment, it is revised.
Line 147 – “epidermal cells” – replace with “of epidermal cells”
[Response]: As your comment, it is revised.
Line 217 - check the font change
[Response]: Following the general nomenclature of Arabidopsis gene (https://www.arabidopsis.org/portals/nomenclature/guidelines.jsp#mutant), mutant gene names and symbols are lowercase and italicized and wild type alleles are uppercase and italicized. The font size you mentioned was fine in Word file. It seems changed during PDF conversion. Its size is double-checked again.
Line 220-221 - check the font change
[Response]: As your comment, it is revised. Its size is double-checked again.
Discussion
Line 229 – “observed by” - replace with “observed thanks to/due to overexpression”
[Response]: Thanks for your kind suggestion. However, we keep ‘by’ instead of ‘due to”. The reason is mentioned above for the Line 2 comment.
Line 265 - comma before "which"
[Response]: Thanks for your right suggestion. As your comment, it is revised.
Line 268 – “by” - - replace with “due to”
[Response]: Thanks for your kind suggestion. However, we keep ‘by’ instead of ‘due to”. The reason is mentioned above for the Line 2 comment.
Line 269 – “indicates” replace with “indicate"
[Response]: As your comment and other reviewer’s comment, it is replaced to “suggest”.
Line 270 – “To be complementary to this theory” - remove "to be"; "complementary with" instead of "complementary to"
[Response]: As your comment, it is revised.
Line 272 - “by” - - replace with “due to”
[Response]: [Response]: Thanks for your kind suggestion. However, we keep ‘by’ instead of ‘due to”. The reason is mentioned above for the Line 2 comment.
Line 277 -“by” - - replace with “due to”
[Response]: [Response]: Thanks for your kind suggestion. However, we keep ‘by’ instead of ‘due to”. The reason is mentioned above for the Line 2 comment.
Line 280 - “by” - - replace with “thanks to”
[Response]: In this sentence, 'by' seems more appropriate than 'thanks to'.
Line 283 – “worth it” - remove "it"; suggestion: "worth exploring" or "worth undertaking"
[Response]: As your comment, it is revised. (Replaced with "worth undertaking")
Line 285 – 286 - check font change
[Response]: As your comment, it is revised.
Line 291 – “all seems to point to microtubules regulated via the regulation of PLD genes are the key” – “all seem to point to the conclusion that microtubules regulated via the modulation of PLD genes are they key factor... (in order to avoid grammar mistake and repetition)”
[Response]: Yes, it is much better. As your comment, it is revised.
References
Line 376 - full stop should be added
[Response]: As your comment, it is revised.
Line 381 - space and full stop should be added
[Response]: As your comment, it is revised.
Line 388 - full stop should be added
[Response]: As your comment, it is revised.
Line 392 - full stop should be added
[Response]: As your comment, it is revised.
Line 398 - full stop should be added
[Response]: As your comment, it is revised.
Line 400 - full stop should be added
[Response]: As your comment, it is revised.
Line 404 - full stop should be added
[Response]: As your comment, it is revised.
Reviewer 2 Report
Comments to manuscript plants-1446495
Interesting and significant work, however several issues should be corrected before publication.
Abstract- it is very difficult to understand the phospholipase-microtubule relationships from the abstract.
Introduction- several aspects presented here are confusing. E.g. what “transversely expanded cell elongation” (Lines/L 77-78) means?
Results
Fig. 2B, detection of YFP signal after plasmolysis- these images are not fully convincing, perhaps an additional fluorescent probe for the detection of cell wall should have been used to see more clearly the distinct localization of plasma membrane and cell wall.
Fig.3- which are the images for Col0 and the overexpressing line respectively?
Discussion
How PLAIIIγ regulates the expression of several MAPs, what could be the mechanism? This should be specified or at least speculated. The paragraph of L 258-283 treats this aspect, but still, I do not really understand the correlations. However the PLA-PLD-MAP link is an interesting suggestion. Please revise this paragraph. This applies to the last paragraph of Discussion as well, where I don’t understand fully the PLD-microtubules-ethylene biosynthesis relationships speculated here by the Authors.
Minor corrections are made in the annotated version of manuscript.

Author Response
Reviewer 2
Interesting and significant work, however several issues should be corrected before publication.
Abstract- it is very difficult to understand the phospholipase-microtubule relationships from the abstract.
[Response]: The abstract part is revised for better understanding by adding the below sentence,
Microtubule-associated protein, MAP18, was enriched in pPLAIIIδ overexpression line in a previous study. Thus, several factors such as microtubules and ethylene-biosynthesis are considered for their radially expanded cell expansion.
Introduction- several aspects presented here are confusing. E.g. what “transversely expanded cell elongation” (Lines/L 77-78) means?
[Response]: As shown in figure 1C, the epidermal cells of pPLAIIIγOE were longitudinally reduced and radially expanded compared to the control. The phrase “transversely expanded cell elongation” indicates the cell elongation characteristics (direction of cell elongation) of pPLAIIIγOE. To have better understanding and as requested by other reviewer too, we have added Figure 8 for “the hypothetical diagram of pPLAIIIg-PLD-MAP-mediated regulation in an anisotropic cell elongation pattern~”. In the diagram of Figure.8B, the defective anisotropic cell expansion is well imaged.
Results
Fig. 2B, detection of YFP signal after plasmolysis- these images are not fully convincing, perhaps an additional fluorescent probe for the detection of cell wall should have been used to see more clearly the distinct localization of plasma membrane and cell wall.
[Response]: In a previous paper (Li et al 2020. Patatin-Related Phospholipase pPLAIIIγ Involved in Osmotic and Salt Tolerance in Arabidopsis. Plants 9(5), 650.), they have also shown that pPLAIIIγ-YFP is localized to the plasma membrane. We re-verified this by merging the fluorescence signals of FM4-64 and pPLAIIIγ-YFP. In that sense, we think that the data in fig 2 is sufficient to understand that pPLAIIIγ-YFP is localized in the plasma membrane.
Fig.3- which are the images for Col0 and the overexpressing line respectively?
[Response]: Upper lane is Col-0 and lower is the OE. The labeling of Figure 3. is revised.
Discussion
How PLAIIIγ regulates the expression of several MAPs, what could be the mechanism? This should be specified or at least speculated. The paragraph of L 258-283 treats this aspect, but still, I do not really understand the correlations. However the PLA-PLD-MAP link is an interesting suggestion. Please revise this paragraph. This applies to the last paragraph of Discussion as well, where I don’t understand fully the PLD-microtubules-ethylene biosynthesis relationships speculated here by the Authors.
[Response]: As your comment, a hypothetical diagram of pPLAIIIg-PLD-MAP link is generated in Figure 8. entitled “A a hypothetical diagram of pPLAIIIg-PLD-MAP-mediated regulation in an anisotropic cell elongation pattern based on transcripts modulation by overexpression of pPLAIIIγ”. Based on our transcriptional changes of PLD and MAP genes by pPLAIIIgOE, we only can speculate and suggest this time. Hope we can support better conclusive remarks in the following work soon.
Minor corrections are made in the annotated version of manuscript.
[Response]: As your comment, it is all properly corrected. Thanks
Reviewer 3 Report
The article is well written, the results are presented quite clearly. However, as always, when studying the interaction of several cellular systems, the final scheme could help the perception. Let some connections be indicated by a dotted line on it, this is quite normal, given that the authors did not study the connections between the players, but only assessed the level of transcription.
It would also be interesting to see the authors' speculations on how exactly the protein under study blocks cell elongation, provoking their isotropic growth. After reading the article, you still don't quite imagine it.
1. The article contains some inaccuracies and unfortunate expressions. For example, "pollen length" (line 111)
2. The title of Figure 1 needs to be changed, it does not fit the meaning.
3. Section 2.3: it seems that in representatives of line No. 14 leaves practically do not grow at all, apparently, not only anisotropic growth is blocked, but also any other one. It would be very interesting to add an illustration with microscopy of leaf cells: to see microparameters in addition to macroparameters (for example, the ratio of the longitudinal to transverse axis for leaf cells). Yet leaf shape and growth pattern are not linearly related.
4. In section 2.4 again there is a mention of mutant lines, although the issue with them was already discussed in 2.1 and it was said that they would not be discussed further.
5. Line 156: "defective in anisotropic cell expansion 156 without altering longitudinal cell growth" - how's that? It is necessary to clarify or say otherwise.
6. Lines 166-174 - speculation, would look better in a discussion, together with an illustrative diagram.
7. Line 205 (and some other) is a rather speculative statement. When the expression of one gene changes in response to a change in the expression of another, this does not mean that one is acting on the other. If the authors made an expression profile, a lot of changes, both direct and indirect, would surely be revealed. In general, this remark applies to all the results mentioned, with the exception of morphometry. You didn’t look at the microtubule structure, you didn’t look at the ethylene level - only PCR. It is quite conditionally possible to draw conclusions about some interactions based on PCR.
8. I would like to see a clear explanation of the differences between the results of this study and those published earlier (5) - it would seem that the two genes are very similar, but such a difference in effects. It is worth at least speculating on this topic.
Author Response
Reviewer 3
The article is well written, the results are presented quite clearly. However, as always, when studying the interaction of several cellular systems, the final scheme could help the perception. Let some connections be indicated by a dotted line on it, this is quite normal, given that the authors did not study the connections between the players, but only assessed the level of transcription.
It would also be interesting to see the authors' speculations on how exactly the protein under study blocks cell elongation, provoking their isotropic growth. After reading the article, you still don't quite imagine it.
[Response]: As your comment, we have added Figure 8 entitled “the hypothetical diagram of pPLAIIIg-PLD-MAP-mediated regulation in an anisotropic cell elongation pattern based on transcripts modulation by overexpression of pPLAIIIγ”. Hope it help better understanding.
- The article contains some inaccuracies and unfortunate expressions. For example, "pollen length" (line 111)
[Response]: Your comment is right. ‘Pollen length’ is changed to “Pollen tube length”
- The title of Figure 1 needs to be changed, it does not fit the meaning.
[Response]: As your comment, it is revised. (Changed the title of Figure 1 to ‘Overexpression of pPLAIIIγ altered cell elongation patterns in floral organs.’). As the overexpression of pPLAIIIγ affected the length of the pollen tube as well as the epidermal cells, the title was changed to inclusive.
- Section 2.3: it seems that in representatives of line No. 14 leaves practically do not grow at all, apparently, not only anisotropic growth is blocked, but also any other one. It would be very interesting to add an illustration with microscopy of leaf cells: to see microparameters in addition to macroparameters (for example, the ratio of the longitudinal to transverse axis for leaf cells). Yet leaf shape and growth pattern are not linearly related.
[Response]: Yes, that's a good point. To better understand the defective anisotropic cell elongation of pPLAIIIgOE, we measured the ratio of the longitudinal to transverse axes using epidermal cells on the axial side of rosette leaves. In OE#14, the ratio was reduced by 52% compared to Col-0 (Col-0 ratio average: 2.91, OE#14 ratio average: 1.41). Relevant sentence is revised under Section 2.3. Added representative epidermal cell images and statistical data in figure 3D, E.
- In section 2.4 again there is a mention of mutant lines, although the issue with them was already discussed in 2.1 and it was said that they would not be discussed further.
[Response]: Yes, you are right. Mentioning mutant lines here again doesn’t make sense. Thus, we deleted the relevant figures and explanation in the main text.
While complete knock-out mutant lines were not available, phenotypic characteristics of two candidate mutant lines, SALK_088404 and SAIL_832_E01, showed similar seed size compared with that of wild-type from two expected lines (Supplementary Figures 1). However,
- Line 156: "defective in anisotropic cell expansion 156 without altering longitudinal cell growth" - how's that? It is necessary to clarify or say otherwise.
[Response]: To better understand, the whole sentence is revised as below,
The exact measurement of the length and width of OE seeds document that the seeds were more transversely expanded defective in anisotropic cell expansion without altering longitudinal cell growth elongation pattern.
- Lines 166-174 - speculation, would look better in a discussion, together with an illustrative diagram.
[Response]: I totally agree with your opinion. Thus, the whole sentence is revised in “Discussion” part as below with simple modification.
They exist in a wide range of species and different proportions, with conversion processes that involve hydrophilic enzymes such as phospholipase-C (PLC), phospholipase-D (PLD), and phospholipase-A (PLA). Phospholipase Dδ (PLDδ) was identified by screening an Arabidopsis cDNA expression library with monoclonal antibody 6G5 against the tobacco 90-kD polypeptide (p90) that was found in microsomal factions and colocalized with cortical microtubules [19, 31]. Cell growth direction usually depends on the organization of the cortical microtubule arrays [32]. It suggests that PLD is involved in signals to the microtubule cytoskeleton. Considering the potential crosstalk between the phosphatidic acid-producing PLD and lysophospholipid-producing PLA pathways in plant-bacterial pathogen interaction [33], the regulatory roles of PLDs by PLAIIIs becomes strongly supported. Transcriptional modulation of PLD genes in pPLAIIIαOE [5] further supports this notion. The PLD and PLA pathways were suggested to crosstalk between the plant-pathogen interactions [29]. Among 12 PLD genes, PLDδ has been characterized for its interaction with microtubules [19,33]. PLDδ associates with the plasma membrane and connects it physically with cortical microtubules [19]. Through this interaction~~
Also, as you suggested, illustrative diagram as Figure 8 is added with proper explanation.
- Line 205 (and some other) is a rather speculative statement. When the expression of one gene changes in response to a change in the expression of another, this does not mean that one is acting on the other. If the authors made an expression profile, a lot of changes, both direct and indirect, would surely be revealed. In general, this remark applies to all the results mentioned, with the exception of morphometry. You didn’t look at the microtubule structure, you didn’t look at the ethylene level - only PCR. It is quite conditionally possible to draw conclusions about some interactions based on PCR.
[Response]: Yes, you are perfectly right. We should have looked the structure of microtubule and ethylene level. We are trying to look at these things in the further study. As your comment, the speculative statement in the last section of 2.6 is deleted as below,
These data strongly indicate that the altered anisotropic cell expansion of pPLAIIIγOE lines was driven by the roles of microtubule-associated proteins, and this regulation is organ-specific through differentially regulated PLD genes (Figure 5).
- I would like to see a clear explanation of the differences between the results of this study and those published earlier (5) - it would seem that the two genes are very similar, but such a difference in effects. It is worth at least speculating on this topic.
[Response]: Basically, the phenotypes by overexpression of all four isoforms of pPLAIII genes are almost similar having a few differences in between. The differences we found is the germination is not changed in pPLAIIIgOE instead of rather faster germination in pPLAIIIaOE.
This part is addressed in the main text as below,
“The consistent germination rate observed in pPLAIIIγOE lines was distinct compared with pPLAIIIαOE that showed an initially increased germination rate [5].”
The novel things observed in this study is the additionally obtained results for example, modulated MAP genes with the reduced transcripts of PLD genes. We will further study whether this change is also common for all pPLAIIIOE or distinct. Of course, it will be more in detail in the further study by dealing in microtubule structure/protein level/ethylene level too.
To have clear idea of the distinct function of each isoforms of pPLAIIIs, we need to further analyzed the change of lipid species (such as ‘lipidomics study’) and substrate specificity of each pPLAIII proteins. It requires more time and efforts. But we are also working on it. Hope we brings more data in the near future.